# Cellular and Molecular Mechanisms of Vestibular Ageing

**DOI:** 10.3390/jcm12175519

**Published:** 2023-08-25

**Authors:** Brahim Tighilet, Christian Chabbert

**Affiliations:** 1Aix Marseille University-CNRS, Laboratory of Cognitive Neurosciences, UMR7291, Team Pathophysiology and Therapy of Vestibular Disorders, 13331 Marseille, France; 2Research Group on Vestibular Pathophysiology, CNRS, Unit GDR2074, 13331 Marseille, France

**Keywords:** neurotology, cellular and molecular ageing, vertigo, dizziness, vestibular disorders

## Abstract

While age-related auditory deficits and cochlear alterations are well described, those affecting the vestibular sensory organs and more broadly the central vestibular pathways are much less documented. Although there is inter-individual heterogeneity in the phenomenon of vestibular ageing, common tissue alterations, such as losses of sensory hair cells or primary and secondary neurons during the ageing process, can be noted. In this review, we document the cellular and molecular processes that occur during ageing in the peripheral and central vestibular system and relate them to the impact of age-related vestibular deficits based on current knowledge.

## 1. Cellular and Molecular Mechanisms of Vestibular Ageing

Ageing is often seen as a pathophysiological process, when in fact, it is a normal process that affects all the body’s tissues and organs. It particularly affects sensory organs and systems, leading to sensory deficits that are perceived in different ways by different individuals. It also modifies a whole range of biological networks and functions, which can affect our balance, our posture and our ability to interact with our environment. Age-related damages to musculoskeletal and tendon networks, visual integration pathways and adaptation mechanisms, as well as metabolic functions and cognitive processes, all contribute to the development of age-related postural-locomotor deficits. In this review, we have chosen to highlight more specifically the age-related alterations that affect the vestibular sensors (crista ampularis, utricles and saccules located in our inner ear) and the central vestibular pathways (with a particular focus on the vestibular nuclei of the brainstem). The data available on this subject in the literature are displayed in the two following sections.

## 2. Ageing of the Peripheral Vestibular System

Recently, a massive age-related loss of synaptic contacts between vestibular hair cells and primary vestibular neurons has been observed using new technological approaches. These observations challenge the established dogmas on the ageing of the vestibular organ by highlighting a major point of fragility within our peripheral sensory organs. Situations in which tissue ageing and its functional correlates can be accelerated must be added to our understanding of the structural degradation of the vestibular system under normal ageing conditions. This is the case, for example, with the acceleration of age-related hearing loss, supported by a drastic decrease in synaptic contacts between inner hair cells and auditory fibres, following auditory over-stimulations demonstrated in young mice. It is possible, although not yet described, that such situations of accelerated sensory ageing may also occur in the vestibule. The morpho-functional observations made in humans post mortem and in animal models of ageing enable us to better understand the natural and pathological ageing of the vestibular organ and to better understand the tissue support of age-related vestibular deficits (Figure 1A).

### 2.1. Age-Related Cell Loss in the Vestibule

Until a few decades ago, the question of the existence of age-related cellular degeneration in the human vestibular system and more widely in mammals was the subject of debate. To date, the question has clearly still not been fully resolved. Almost 50 years ago, Rosenhall was the first to mention the lack of a clear correlation between vestibular deficits and tissue and cellular alterations [3]. In the following decades, the process of age-related degeneration was described in humans in almost all cell types considered [4,5,6,7,8,9,10]. This lack of a direct correlation between cell loss and vestibular deficits is now interpreted as the consequence of the combined actions of central vestibular compensation [11] and peripheral synaptic repair [12]. The conjunction of these two reactive processes masks the functional consequences of progressive degeneration, which is nevertheless present in sensory tissues [7,9,10]. The majority of histological studies examining vestibular hair cell degeneration confirm a significant onset of hair cell decline from the age of 65–70 years. The first histopathological analyses, carried out as early as 1975 by Rosenhall and Rubin, concluded that the number of hair cells in the vestibular organs remained relatively stable up to the age of 70, before declining significantly [13]. However, they reported a significant and marked reduction in the number of hair cells in the crista of the semicircular canals from the age of 50 to 60 years. A significant reduction in the number of neurons in Scarpa’s ganglion was observed as early as 60 years of age [14], while a significant reduction in vestibular nerve fibres began as early as 50 years of age, with losses of up to 40% in individuals aged 75 to 80 years [15].

Using Nomarski interference contrast microscopy, Merchant and colleagues determined the total number of vestibular hair cells in each of the five vestibular organs (the crests of the three ampullae of the semicircular canals and the utricular and saccular maculae) [1] (Figure 1B). From a series of 67 normal temporal bones collected from birth to 100 years of age, the authors identified a strong age-related decrease in vestibular hair cells in the different types of sensory epithelia, with a significantly greater loss of stereocilia in the cristae than in the maculae. They did not find a significant difference in age-related cell loss between the three semicircular canals, the two maculae, males and females, or the contralateral ears. Velazquez-Villasenor et al. proposed that a linear decrease in Scarpa’s ganglion primary vestibular neurons takes place from birth to the most advanced stages [16] (Figure 1C). Rosenhall reported losses of sensory cells and primary neurons as early as 40 years of age, culminating in a 40% loss of vestibular sensory cells by 75 years of age [3]. A greater susceptibility to age-related cell loss in the semicircular canal cristae (40% hair cell loss) compared to that in the maculae (20–25% hair cell loss) has also been described, and this difference is accentuated in the 70–95-year age group compared to the 40-year age group [3,4] (Figure 1D).

In summary, the loss of hair cells and primary vestibular neurons occurs in men from the age of 40 years, becoming significant in the period 70–90 years. During this period, tissue damage is more significant in the ampullae than in the maculae sensory epithelia. In his review on the ageing of the human vestibular system, Zalewski indicated that cell loss affects the centre of the sensory epithelia more than the periphery and that type 1 hair cells (those afferent to calyx nerve endings) are more fragile than type 2 cells [17]. Within the otolithic organs, cell loss is higher in the saccule than in the utricle. Age-related cell loss is also seen in Scarpa’s ganglion.

### 2.2. Age-Related Synaptic Loss and Vestibular Ageing

Recently, using a novel histological approach combining an immunocytochemical analysis and quantification of synaptic proteins, the team of Prof. Corfas from the Kresge Hearing Research Institute at the University of Michigan in Ann Arbor (USA) has made a major step forward in our understanding of the cellular mechanisms impacted by vestibular ageing [18]. The team compared the number of synaptic contacts between vestibular hair cells and the nerve terminals of the fibres that form the vestibular nerve in mice aged 2, 5, 9, 18 and 24 months. These different observation ages corresponded approximately to the periods of being a young adult (20–30 years), mature adult (40 years) and older adult (65–75 years) in humans. For this purpose, using fluorescent antibodies, the authors labelled the CtBP2-Ribeye protein (presynaptic protein expressed at the level of the docking zone of synaptic vesicles in the basolateral part of hair cells) on the one hand and the GluA2 protein (protein subunit of AMPA-type glutamate receptors expressed at the post-synaptic level in the afferent terminals of primary vestibular neurons) on the other hand. Using other associated markers, such as tenascin C (a protein specific to calyx terminals), secreted phosphoprotein 1 (a protein specific to type 1 cells) or myosin VIIA (a protein of ciliary tufts), the authors succeeded in analysing the variations in the expression kinetics of these proteins, as an indicator of the integrity of the two types of synapses (bouton and calyx), according to their location on the sensory epithelium (striolar versus extrastriolar) over time (Figure 2). Their analyses demonstrated that, over the period considered, the number of hair cells, regardless of their type or the area considered, did not vary significantly between the young and very old stages (Figure 2A–D). Similarly, the number of neurons in Scarpa’s ganglion did not vary significantly between the different stages (Figure 2E,F). In contrast, the analysis of the temporal variation in synaptic protein expression revealed a significant decrease in pre-and post-synaptic proteins only at the oldest stage. This synaptic damage specifically affected calyx synapses preferentially located in the periphery of sensory epithelia in the so-called extra-striolar zone (Figure 2H–K). In order to study the functional consequences of these tissue alterations, the authors recorded the vestibular evoked potentials (VEPs) of mice of the same type and at the same ages as for the histological analyses. Their observations showed that, in mice, the deterioration of VEPs only appeared at the 24-month stage (Figure 2G). Thus, the age-related impairment of VEPs is significantly correlated with synaptic loss and not with the loss of hair cells or primary vestibular neurons. These results demonstrated that the loss of extra-striolar calyx synapses has a key role in age-related vestibular dysfunction (Figure 2L). On this basis, studies on the mechanisms of synaptic plasticity in excitotoxic injury [19,20], which have revealed a strong potential for post-injury reafferentation after selective hair cell deafferentation, may provide a real opportunity to discover new pharmacological pathways to slow down the ageing of vestibular synapses or even stimulate their repair. This type of pharmacological approach could constitute, together with vestibular rehabilitation by physiotherapy, future interventional therapies.

### 2.3. Conditions That Accelerate Age-Related Synaptic Loss

A significant acceleration of age-related auditory primary neuron loss was previously demonstrated in mice, secondary to auditory over-stimulation early in the animals’ lives [21]. In this model, auditory trauma led immediately to a loss of synaptic contacts between the inner cells and spiral neurons, without affecting the hair cells or spiral ganglion neurons themselves. In contrast, observations of the temporal bones of these same mice at later stages revealed a very significant acceleration of age-related losses of auditory neurons. At the vestibular level, on the basis of the observations made in the cochlea, it cannot be discarded that peripheral synaptopathies occurring at the young adult stage could also accelerate the age-related loss of primary vestibular neurons and aggravate their functional consequences.

### 2.4. Spontaneous Endogenous Mechanisms to Slow down Age-Related Synaptic Loss

The spontaneous reafferentation of inner ear hair cells is one of the hypotheses to explain the functional restoration following acute peripheral vestibulopathies, such as vestibular neuritis or labyrinthitis [22]. In order to investigate the reality of such a process, the Physiopathology and Therapy of Vestibular Disorders team at UMR7291 in Marseille developed in vitro and in vivo models of selective damage to inner ear synapses [19,20,23,24,25]. The first co-culture model of rodent vestibular endorgans and Scarpa’s ganglion demonstrated that the reafferentation process was composed of different phases and that it resulted from trophic interactions between the regrowing fibre and its target (Figure 3A–D) [24]. Another in vivo study model allowing the induction of selective lesions of vestibular primary synapses (through a transtympanic injection of glutamate receptor agonists) has been developed in rats and mice. This study model has enabled us, on the one hand, to examine the correlation between the severity of peripheral damage and the various symptoms that compose posturo-locomotor syndrome [20,25] and on the other hand, to analyse the rearrangement of synaptic proteins, such as CtBP2 and Shank-1 (one of the components of the protein scaffold found at the post-synaptic terminals) during the deafferentation and reafferentation process (Figure 3E–K) [19,20]. These different studies combining in vitro and in vivo approaches have thus confirmed the existence of a spontaneous synaptic repair phenomenon within mammalian vestibular organs. This process could contribute to slowing down age-related synaptic loss and in turn slowing down age-related vestibular deficits.

### 2.5. Current Data on Ageing Affecting Otoconia Metabolism

Other components of vestibular sensors may be affected by aging. However, even though several studies have identified the genes and molecular processes involved in the birth and formation of otoconia [26,27], there is still very little documentation on the fate of otoconia during normal ageing. Under normal conditions, the saccule and utricle are covered by a dense gelatinous layer of protein filaments (called otoconial gel), which holds the otoconia together and prevents their dispersion [28,29]. An alteration in these protein filaments during ageing has been reported specifically in the saccule [30,31]. This phenomenon is likely to favour the detachment of otoconia from the otoconial membrane. A progressive reduction in the number of otoconia from 50 years old onwards has been reported with a preponderance in the saccule [32,33]. The displacement of whole or fragmented otoconia can cause or contribute to dysfunction of the crista ampullaris in the form of cannalolithiasis. These particles may alter the movement of the cupula and thus affect mechano-electrical transduction. One reason for this may be the vertical orientation of the saccule in the temporal bone, making it more susceptible to otoconia loss. It has also been suggested that the loss of otoconia may be secondary to the loss of dark cells in the saccule [29]. A strong correlation between benign paroxysmal positional vertigo (BPPV) and ageing has been demonstrated in women [34]. This correlation may have its source in the changes in the conformation of the otoconia following alterations in the secretion of sex hormones, as has been demonstrated in a rodent model [35].

### 2.6. Functional Consequences of Age-Related Damages to the Peripheral Vestibular System

It is still challenging to establish a precise correlation between the age-related cellular and molecular changes mentioned above, affecting the different components of vestibular sensors, and age-related balance disorders. Several excellent recently published reviews, combining observations in humans and animal models of ageing, have reported the current evidence for age-related changes in the vestibular system and their importance to clinicians treating balance disorders [36,37].

## 3. Ageing of Central Vestibular System

Like the peripheral vestibular system, the central vestibular system displays a remarkable capacity for self-repair. Vestibular compensation, a model of post-injury plasticity in the central nervous system, refers to a set of endogenous mechanisms of neuroplasticity, which are well described in the vestibular nuclei, in response to damage to the peripheral vestibular system and underlying functional restoration. In the course of ageing, this ‘homeostatic’ plasticity, although still present, diminishes and is accompanied by sensorimotor and cognitive disturbances. Independently of age, vestibular compensation can be improved by pharmacological therapy and by physical rehabilitation based on the strengthening of other sensory modalities, such as vision or proprioception, and of cognitive and motor components. In the following paragraphs, we first describe the neurobiological mechanisms of vestibular compensation and then discuss the impact of ageing on this adaptive plasticity.

### 3.1. Central Integration of Vestibular Information: The Vestibular Nuclei, a Central Information Processing Hub

Following its encoding in the inner ear vestibular sensory organs, the vestibular sensory information travels along the vestibular nerve (the eighth cranial nerve) to a first central integration relay, the brainstem complex of the four vestibular nuclei (medial, lateral, inferior and superior), which transforms it into pre-motor, vegetative, perceptual and cognitive information [38]. Neurons of the vestibular nuclei project their axons to motor neurons located at all levels of the brain and spinal cord, which innervate the oculomotor muscles, muscles of the abdomen and back, as well as those of the upper and lower limbs. These networks are involved in stabilizing the gaze and controlling one’s posture at rest and during movement through the vestibulo-ocular, vestibulo-cervical and vestibulo-spinal reflexes. The vestibular nuclei also transmit the vestibular sensory information to a set of brain areas involved in perceptual and cognitive functions, such as in the appreciation of the verticality of the body and its orientation in space [39]. The interconnection of vestibular nuclei with structures of the autonomic nervous system constitutes the first relay of the vestibulo-vegetative reflexes responsible for the discomfort (such as nausea) felt in vestibular disorders [40] (Figure 4A). The vestibular nuclei do not exclusively process vestibular sensory information; they behave as a ‘hub’ [41] as they are interconnected to several networked nerve structures, allowing information from other sensory modalities (proprioceptive, tactile and visual [41,42]) to be combined. This information also contributes to postural, oculomotor, perceptual and cognitive functions.

### 3.2. Vestibular Syndrome: A Range of Oculomotor, Postural, Perceptual, Cognitive and Vegetative Symptoms

Unilateral damage to the vestibular sensors induces a characteristic acute vestibular syndrome in various species, including humans. This syndrome is the result of the alteration in the vestibulo-spinal, vestibulo-oculomotor and vestibulo-vegetative reflexes, as well as the alteration in vestibulo-cortical signals [38,43]. It consists of postural imbalance at rest and during movement as well as the loss of coordination of eye movements in the orbits (nystagmus, oscillopsia) and is associated with perceptual, cognitive and neurovegetative disorders (Figure 4B). This characteristic phenotype is also observed in unilateral peripheral vestibular impairments resulting from vestibular neuritis, Meniere’s disease or labyrinthine fistula [41]. In addition, patients with vestibular disorders with episodic vertigo, such as those encountered in vestibular migraines and Meniere’s disease, suffer from anxiety and affective disorders [44]. This emotional component of the vestibular syndrome is explained by the neural substrates linking balance control and anxiety [45]. Over time, the clinical symptoms diminish, which usually leads to an almost complete disappearance of the syndrome. This phenomenon of spontaneous behavioural recovery is called ‘vestibular compensation’ [46].

In most species, post-injury vestibular syndrome includes static deficits (present when the body is not moving) and dynamic deficits (present during displacements). Static deficits include oculomotor (spontaneous nystagmus) and postural (head tilt and postural instability) alterations, which are compensated for within a few days or weeks. Dynamic deficits (vestibulo-ocular and vestibulo-spinal reflexes and locomotor performance), on the other hand, are compensated much less completely and over a longer period of time [47,48].

### 3.3. Vestibular Compensation upon Ageing

The vestibular compensation process involves neurochemical, hormonal, structural, metabolic and electrophysiological reorganizations within the vestibular nuclei disconnected from their peripheral sensors. During vestibular compensation, the loss of information from the vestibular sensory organs leads to a reweighting (or sensory substitution) of the different sensory channels (visual, somatosensory and vestibular) in order to maintain the balance function [49,50].

Brain ageing is characterized by several neurochemical changes involving structural proteins, energy metabolism, neurotransmitters, and neuropeptides and their receptors [51]. Age-related changes in the somatosensory, visual and vestibular systems have a negative impact on postural balance, which increases the risk of falls in affected individuals. The sensory systems used by the vestibular centres to optimize compensation for peripheral vestibular damage are therefore also impacted by ageing, which leads to a generalized decline in sensory functions and in the capacity of the central nervous system to integrate sensory information, decreasing the gain for these different modalities [52]. The endogenous mechanisms of neuroplasticity in the vestibular nuclei and the sensory substitution strategies put in place to optimize vestibular compensation are therefore altered by ageing.

Like sensory modalities, the motor component, represented by the musculoskeletal system, plays a key role in the maintenance of balance in both young and older adults. However, this system is also altered by ageing. The age-related loss of muscle function involves quantitative and qualitative changes in skeletal muscle structure and function. This process is generally slow, and the loss of function varies considerably from one individual to another, but it occurs in all individuals, even those who are healthy, well-nourished and physically active. The decrease in muscle mass and the accompanying loss of function is referred to as sarcopenia. It is one of the most apparent changes in the aging process [53]. Given the impairment of the sensory and motor systems in older adults, it is to be expected that vestibular compensation is less effective in these individuals. However, there is little research on the quality of vestibular compensation and the efficiency of its mechanisms during ageing in humans and animals. Some events that occur in the vestibular nuclei during ageing could nevertheless explain why vestibular compensation is poor in these older adults.

### 3.4. Anatomical Changes in the Vestibular Nuclei during Ageing

The data regarding neuronal density in the vestibular nuclei during ageing are rather controversial. Neuronal losses have been observed in the vestibular nuclei complex in humans [54,55,56,57] and in animals [58]. In humans, the number of neurons forming this complex decreases by about 3% per decade from the age of 40 onwards [24], but no significant age-related decrease in the number of neurons has been found in golden hamsters [59]. There is also a reduction in the number of cerebellar Purkinje cells that contribute to the modulation of central vestibular excitability necessary for the management of postural and locomotor balance, due to their projection onto the vestibular nuclei [60]. Spinal motor neurons are also affected by ageing. The reduction in their number is accompanied by more activated levels of microglia. These data suggest that the loss of spinal motor neurons in older adults may contribute, in part, to late motor impairment [61].

Johnson and Miquel [62] analysed the ultrastructure of the lateral vestibular nucleus of rats at 4 weeks, 6–8 weeks, 6–8 months and 18–20 months of age. Let us recall the crucial role of the lateral vestibular nucleus in postural-locomotor balance. In these animals, the authors observed certain structural alterations of the cells, the frequency of which increased with age: the presence of invaginations of the nuclear membrane, a disorganized endoplasmic reticulum, rod-shaped nuclear inclusions, and dense cytoplasmic bodies of the lipofuscin type, a typical tissue marker of ageing. The oldest group of animals also showed axonal degeneration and the swelling of dendrites. Dystrophic axonal terminals have also been reported in the vestibular nuclei of one-year-old gerbils [63]. The neurons’ cytoplasm showed neurofilaments and vesicles with membranous granular substances. These data demonstrated the existence of the age-related structural deterioration of neuronal networks in the vestibular nuclei, apart from neuronal loss.

The apparent discrepancies between the results of different functional and structural studies in aged animals and humans suggest considerable variability in the effects of ageing on the vestibular system. In addition to a probable interspecies difference, it is likely that much of the variability within a species is due to genetic and epigenetic differences influencing the adaptive plasticity of the ageing vestibular system.

### 3.5. Neurochemical Changes in the Vestibular Nuclei during Ageing

It is reasonable to assume that age-related alterations in vestibular function result in part from neurochemical changes in the neural circuitry involved. To date, most studies in aged animals have been limited to the brainstem vestibular nuclei. The vestibular nuclei are connected in a network of several nerve structures, allowing them to receive and integrate information from different sensory modalities for transmission to other structures. These axons, which converge on the vestibular nuclei, release a wide variety of neurotransmitters. The main afferents to the vestibular nuclei come from the vestibular primary neurons, whose neurotransmitter is glutamate. Neurons in the vestibular nuclei also receive glutamatergic afferents from the spinal cord, GABAergic afferents from the cerebellum and contralateral vestibular nuclei via the commissural system, dopaminergic afferents from the midbrain nuclei, serotoninergic afferents from the raphe nuclei, and histaminergic afferents exclusively from the tuberomammillary nuclei of the hypothalamus. Figure 5 shows a map of the neurochemistry of the central vestibular nuclei. Below we report age-related changes affecting these different components.

#### 3.5.1. Changes in the Glutamatergic and GABAergic Systems

Glutamate and GABA are neurotransmitters associated with the excitation and inhibition of neurons in the central nervous system, respectively. They function in synchrony to maintain an excitation/inhibition balance (so-called electrophysiological homeostasis). Vestibular neuronal networks function predominantly with these two neurotransmitters. The electrophysiological responses of medial vestibular nucleus neurons to NMDA (N-methyl-D-aspartate), AMPA (aminomethylphosphonic acid) and kainate (agonists of the three types of ionotropic glutamate receptors), recorded in vitro on brainstem slices, are similar in young (3 months) and old (24 months) rats, suggesting that there is no change in the sensitivity of the three ionotropic glutamate receptor subtypes to their selective agonists during ageing [64]. A comparison of glutamate concentrations in the vestibular nuclei complex and cerebellum of 4-, 12- and 24-month-old rats using high-performance liquid chromatography (HPLC) revealed a significant decrease in this neurotransmitter in the vestibular nuclei complex and cerebellum as a function of animal age [65,66]. The results of these two types of studies [65,66,67] are not necessarily inconsistent. It is indeed possible that the number of AMPA and NMDA receptors, or their sensitivity to glutamate, increased in response to the decrease in neurotransmitter concentration in the vestibular nuclei complex, which could then lead to a neuronal response to these different agonists that would be little altered during ageing.

Him et al. studied neurons in the medial vestibular nucleus of 24-month-old rats and found an increased sensitivity to the GABAa receptor agonist muscimol. They concluded that this was a compensatory response to the age-related loss of neurons in the medial vestibular nucleus [67]. Giardino et al. also assessed the levels of mRNA specific to glutamic acid decarboxylase (GAD), the enzyme for GABA synthesis, in the vestibular nuclei complex of 24-month-old rats. Observing increased levels of mRNA, they concluded that there was an increased synthesis of GABA in the aged animals [68]. However, these results remain controversial, as Liu et al., using homogenized brain samples and the HPLC technique, observed no significant changes in GABA concentrations in the vestibular nuclei complex and cerebellum in rats during ageing [65,66]. To date, few studies have investigated age-related changes in glycine receptors in the vestibular nucleus complex. However, Nakayama et al. showed a strong age-dependent decrease (3, 18 and 26 months) in the binding of strychnine, a specific antagonist of these receptors, in the vestibular nuclei complex. The amount of strychnine that binds to the cells decreased by half between 3 and 26 months [69]. The functional significance of this decrease in glycine receptor density during aging remains unknown. However, the authors hypothesized that increased glycine synthesis during ageing could prevent an imbalance between neuronal excitation and inhibition. Recent studies in rats aged 6, 22 and 33 months have shown significant decreases in the concentrations of glutamate, aspartate, glycine, GABA and taurine in the vestibular nuclei complex with age. Conversely, glutamine, threonine and serine concentrations are significantly increased with age. In contrast, choline acetyltransferase concentrations do not appear to be altered, suggesting no change in acetylcholine availability [70].

#### 3.5.2. Changes in Monoaminergic Systems

Using brain tissue samples and the HPLC technique to measure the concentrations of norepinephrine (NA), serotonin (5-HT) and dopamine (DA), as well as their metabolites in the medial vestibular nucleus of rats aged 4, 21 and 24 months, Cransac et al. observed that NA concentrations decreased with age, while those of 5-HT and its metabolite, 5-hydroxyindoleacetic acid (5-HIAA), increased. In contrast, the levels of DA and its metabolite, 3,4-dihydroxyphenylacetic acid (DOPAC), remained unchanged [71]. Di Mauro et al. have suggested that the decrease in NA in the vestibular nuclei complex may contribute to the age-related deterioration of vestibular function, although the cellular and molecular mechanism remains enigmatic [72].

### 3.6. Other Changes in Neurobiological Factors during Aging That Are Essential for Vestibular Compensation

Neurogenesis, the level of neuronal excitability, BDNF (brain-derived neurotrophic factor), inflammation players (astrocytes and microglia), stress hormones and neurochemical systems (GABA, acetylcholine and histamine), all of which are elements of the deafferented vestibular environment important for vestibular compensation [9], also undergo alterations during ageing. In rodents, in the two main germinal nerve centres of the brain, the subventricular zone of the lateral ventricles and the subgranular zone of the dentate gyrus of the hippocampus, there is a decline in the production of new neurons during ageing [73] that has been linked to the functional loss of olfaction [74] and to the decline of hippocampus-dependent spatial memory [73], respectively. Yet, the new neurons that are produced in the hippocampus appear to be functionally equivalent to those present in the young brain [75]. Neurogenesis in the aged brain is therefore not aberrant. It is simply less active than in the young brain. A recent study carried out in older adults confirms these observations obtained in animals concerning the reduction in the formation of new hippocampal neurons associated with a decline in spatial memory during ageing. Combining analyses of hippocampal volume and shape, in particular to better understand the relationship between vestibular function and this brain structure in an ageing population, an association between reduced hippocampal volumes and poor vestibular function was observed [76].

Old age is characterized by increased expression levels of pro-inflammatory cytokines and decreased expression of genes whose products are anti-inflammatory, resulting in a chronic low-grade inflammatory state in the brain. In the aged brain, microglia are more active. Faced with a pro-inflammatory stimulus (such as bacteria), these cells are hyper-reactive and produce greater quantities of cytokines and over a longer period of time. This could explain the sometimes-prolonged behavioural disturbances in older adults, which may be accompanied by cognitive impairment, in the event of infection or other acute illnesses, and could also contribute to the development of age-related neurodegenerative diseases [77]. Ageing also leads to the dysfunction of astrocytes, which reduces their ability to maintain a healthy environment for neurons. This also alters their interactions with neighbouring cells and contributes to the low-grade inflammatory state characteristic of ageing [78].

Brain-derived neurotrophic factor (BDNF) is associated with neuronal maintenance, neuronal survival, plasticity and neurotransmitter regulation. Although its expression in the brain does not appear to be altered with age, some studies have shown a reduced ability of the older adult brain to produce this factor in response to stress [79]. Patients suffering from psychiatric disorders or neurodegenerative diseases often have reduced concentrations of BDNF in the blood and brain. These abnormally low levels of BDNF could be due, at least in some cases, to a chronic inflammatory state. Neuroinflammation, resulting from this chronic inflammatory state, is indeed known to affect several signaling pathways that are linked to BDNF [80].

The plasticity of intrinsic neuronal excitability facilitates learning and memory processes. Reductions in intrinsic excitability, as well as aberrant plasticity and, in some cases, pathological hyperexcitability, have been associated with cognitive deficits in animal models of cognitive ageing or Alzheimer’s disease. Targeting molecular mediators that control intrinsic neuronal excitability, which would promote learning-related intrinsic neuronal plasticity, is a promising therapeutic strategy for maintaining cognitive function during ageing or in response to neurodegenerative disease [81].

It therefore seems clear that aging alters the neurobiological plasticity mechanisms essential for restoring and maintaining vestibular function in the event of damage. Aging of vestibular function, with its consequences on the elderly’s postural instability and frequency of falls, is likely to result from an alteration in these same plasticity mechanisms within peripheral and central vestibular structures. Specifically targeting these different endogenous plasticity mechanisms through pharmacology and/or rehabilitation could thus be a promising therapeutic strategy for maintaining vestibular function during aging.

### 3.7. Example of Vestibular Compensation in Older Adults and Animals

This is a very poorly documented area. There are few data on the different vestibular symptoms and their evolution with age. In our laboratory, we have developed a mouse model of unilateral vestibular injury induced by excitotoxicity in order to compare the ability of young (3 months) and old (22 months) adult mice to restore their balance and posture [25]. In this model, we followed the progression of the vestibular syndrome over time, using a battery of behavioural tests to monitor the evolution of postural and locomotor abnormalities. We observed that young and old adult mice recovered normal postural-locomotor function after a transient unilateral excitotoxic vestibular lesion. However, the kinetics differed. The restoration of postural-locomotor and swimming balance was significantly slower in the aged mice. This original work thus opens the way to more detailed studies on the plasticity mechanisms responsible for the restoration of postural-locomotor function after unilateral peripheral vestibular damage in older adults, in particular on the impact of ageing on these mechanisms. In an earlier study, the temporal changes in vestibular compensation in young (3 months) and old (24 months) adult rats were examined. A correlation between the expression of the gene coding for glutamic acid decarboxylase (GAD), the enzyme for the synthesis of GABA and the density of benzodiazepine receptors in different brain areas, including the vestibular nuclei, was established. After the unilateral destruction of the peripheral vestibular organs (hemilabyrinthectomy), compensation in the young adults was complete after 28 days, whereas the old rats still showed significant behavioural disturbances. In the latter, a higher GABAergic tone was observed, resulting in a higher density of benzodiazepine receptors in the lateral vestibular nucleus and a higher level of GAD-specific mRNA in the cerebral cortex and medial vestibular nucleus. In contrast, in young adults, the benzodiazepine receptor density was normal in the vestibular nuclei at 28 days post-lesion, and the DSM mRNA levels were higher in the anterior cingulate cortex only. In aged rats, however, these parameters were still altered in the anterior cingulate cortex, somatosensory cortex, basal ganglia, vestibular nuclei and cerebellum 28 days after the vestibular injury [64]. Alterations in their GABAergic transmission and behavioural profile after unilateral peripheral vestibular injury are therefore impacted by ageing. The effects of acute unilateral peripheral vestibular loss, presumably due to vestibular neuritis, were examined by Scheltinga et al. in young (23–35 years), middle-aged (43–58 years) and older (60–74 years) adult subjects [82]. In these individuals, the results indicated that acute unilateral vestibular damage causes a relative worsening of postural control and gait balance in the older adults compared to young adults. In the acute stage, the older adults were more unstable than the young adults and took longer to recover the balance abilities of age-matched controls.

### 3.8. Re-Emergence of a Critical Post-Injury Period Reproducing Developmental Plasticity Processes

The mechanisms of neuroplasticity in vestibular nuclei disconnected from their peripheral afferents occur within a time window, between 3 and 30 days after unilateral vestibular nerve transection in different species. The first four weeks after unilateral peripheral vestibular injury probably represent a critical period, during which these adaptive neuroplasticity mechanisms, including vestibular neurogenesis, are activated to promote vestibular functional recovery. In support of this notion, a sensorimotor restriction applied during the first week after a unilateral vestibular nerve transection in an adult cat or monkey prevents postural recovery, whereas when applied later, this restriction no longer has an effect on their recovery [83,84]. The re-emergence of a critical period during the post-injury self-repair phase is reminiscent of certain scenarios in brain development, in which neurogenesis, the maximal re-expression of neurotrophins (BDNF) and GABA (then excitatory) are solicited simultaneously [85]. If the aging of the vestibular system can indeed be considered as a non-pathological process, the peripheral and central tissue lesions it causes could activate the same reactive and adaptive cellular and molecular mechanisms as those encountered after vestibular injury. In the same way, vestibular injury in an aging person could also reproduce the same developmental scenario for restoring and maintaining vestibular function.

Interestingly, the molecular ingredients used by the mature deafferented vestibular environment (EGR1, Fos, BDNF, CamKIIa (Calcium/calmodulin-dependent protein kinase II)) for the compensation process are similar to those used in the hippocampus for the induction of the electrophysiological phenomenon of long-term potentiation, which probably represents the neurobiological substrate of learning and memory [86]. This potentiation phenomenon is also present in intact vestibular nuclei [87,88], but also in deafferented vestibular nuclei after a peripheral vestibular lesion [89]. We can therefore postulate that the vestibular compensation phenomenon is comparable to a sensorimotor relearning process, using the same mechanisms as those used in the hippocampus. The vestibular nuclei are closely linked to the hippocampus [90]. From a behavioural point of view, the acute postural-locomotor phenotype induced after a unilateral peripheral vestibular lesion is reminiscent of learning and developmental strategies for walking [91]. This suggests that the compensation of walking and postural balance following such an injury can be likened to sensory and motor relearning. The adaptive process of vestibular compensation can be likened to a form of cellular or behavioural resilience. Resilience originally referred to the resistance of a material to shocks, a definition extended to the capacity of a body, an organism, a species, a system or a structure to overcome an alteration in its environment. In psychology, resilience refers to the ability of individuals who have suffered a traumatic event to draw on their psychological resources in order to rebuild himself or herself in a socially satisfactory way, by implementing adaptive strategies. We can postulate that a vestibular trauma occurring in older adults reactivates endogenous plasticity processes, such as those used during development (cellular and molecular mechanisms, sensory substitution strategies, sensorimotor relearning processes) and those that have contributed to the maturation of the balance function. This reactionary state, which is specific to each individual, is equivalent to a reset of the vestibular function, restoring a new form of balance in older adults, enabling them to provide adaptive responses to daily demands. The vestibular syndrome caused by a trauma is accompanied by a stress component which is sometimes very anxiety-provoking for the individual and therefore deleterious for vestibular compensation in older adults, who are known to be vulnerable to stress. Regarding vestibular pathology in older adults, psychotherapy can therefore contribute to the reconstruction of vestibular function, as a complement to medical treatment and physical rehabilitation. BDNF plays a crucial role in normal vestibular function and post-traumatic plasticity. Through its receptor associated with tyrosine kinase B (TrkB), it participates in the construction of the vestibular system during development [92]. It is highly re-expressed in vestibular nuclei under pathological conditions [93] and promotes vestibular compensation in adult mammals [94]. It is found in almost all neuroplasticity processes promoting vestibular compensation: neurogenesis, synaptic plasticity, the restoration of a homeostatic neuronal excitability level, etc. It is interesting to mention that the expression of this growth factor is stimulated by physical activity [95], a balanced diet [96] and by various meditation techniques [97]. These different practices can therefore prevent the deterioration of the vestibular system in older adults and improve their quality of life.

### 3.9. Future Insights into the Study of Vestibular Aging

The correlation between age-related cellular and molecular damage and the posture and balance deficits observed in older adult patients, as well as in animal models of ageing, leads us to think that we have identified the mechanisms that support vestibular ageing. However, we are still far from understanding how age impacts the plasticity properties that make the vestibular system unique, such as the repair of peripheral vestibular synapses or central compensation. However, this is a challenge that must be met if we are to understand how vestibular damage in young adults governs the subsequent evolution of vestibular sensory networks. The spread and simplification of biotechnology approaches suggests that there is scope for interventions in the vestibular sphere to protect or repair sensory cells, their contacts with the fibres that form the vestibular nerve, or the central networks involved in the integration of vestibular sensory information. It is true that approaches involving neurotrophins, for example, may hold out the hope of being able to slow down the action of time on our tissues. However, any interventional action must first be based on a real understanding of the critical periods present during the development of the vestibular sensory networks, but which are also found throughout life in the repair or compensation processes in situations of vestibular damage. Methods of the clinical exploration of vestibular function will also have to evolve to allow for more precise and specific monitoring of the changes in the sensory organs and vestibular networks during ageing. These data will allow for the better targeting of preventive and therapeutic approaches.

## 4. Conclusions

Throughout this review, we mentioned divergences between studies in animals and in humans; however, we must remember that the human body in relation to balance requires more coordination and motor control compared to animals, due to the bipedal position adopted by humans. This leads to a smaller base of support and consequently a greater intensity of muscle co-contractions to maintain a desired posture or maintain one’s posture without causing changes, pain or damage to the individual. This needs to be considered in the weighting between animal and human studies.

In this review, we have chosen to shed light specifically on age-dependent alterations which affect both the vestibular sensors and central vestibular pathways. It is obvious that these age-dependent tissue damages, together with the loss of neural plasticity processes, are one of the major causes of balance and posturo-locomotor deficits. We know, however, that the efficacy and reactivity of the gate and balance processes also result from the integration of inputs from vision and proprioception and that the functional state of the vestibulo-ocular reflex and vestibulo-spinal pathways is critical for good coordination of one’s equilibrium both at rest and during displacements. The processes of body self-perception and the orientation of our body in space are also essential parameters to our interaction with our environment [98] and for the precise control of our gait and balance. Therefore, age-related deficits affecting the musculoskeletal and tendon networks, visual integration pathways and adaptation mechanisms, metabolic functions and cognitive processes must be taken into account in interventional actions such as vestibular rehabilitation in aged populations.

## Figures and Tables

**Figure 1 jcm-12-05519-f001:**
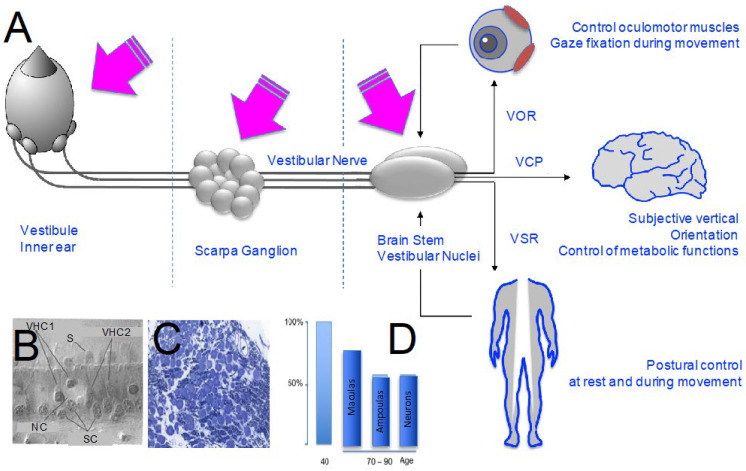
Age-related cell loss in the vestibular system. (**A**) Illustration of areas impacted by ageing in the vestibular system in humans and mammals. VOR: vestibulo-ocular reflex; VSR: vestibulo-spinal reflex; VCP: vestibulo-cortical pathways. (**B**–**D**) Histological analysis of age-related cell loss in the vestibule. (**B**) Utricle of a 54-year-old man displaying vestibular hair cells 1 and 2 (VHC1-VHC2), stereocilia (S), nerve calyx (NC) and supporting cells (SC) [1]. (**C**) Slice of Scarpa’s ganglion from an adult rat displaying vestibular primary neurons cell bodies [2]. (**D**) Diagram of age-related cell loss in cristae, maculae and Scarpa’s ganglion [3,4].

**Figure 2 jcm-12-05519-f002:**
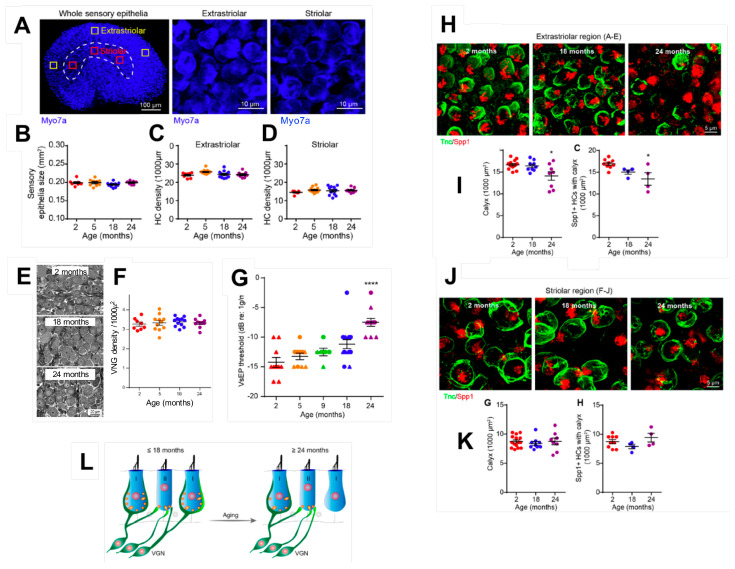
Histological analysis of age-related synaptic loss in mice. (**A**) Images of mouse utricles with hair cell labelling (myosin VIIA) in striolar and extrastriolar areas. (**B**–**D**) Age-dependent counts of utricle size and hair cell density in mice that are 2, 5, 18 and 24 months old. (**E**,**F**) Image and density of primary neurons in Scarpa’s ganglion at different developmental ages. (**G**) Vestibular evoked potentials at different ages (**H**–**K**). Images and counts of calyx endings as a function of age in different areas of the sensory epithelium. Immunocytochemical labelling of tenascin-C (green) and secreted phosphoprotein 1 (red). (**L**) Illustration of the process of age-related synaptic loss. Vestibular ganglion neuron (VGN). * *p* < 0.05 and **** *p* < 0.0001 by one-way ANOVA comparing to 2-month-old mice followed by Dunn’s post-test. See [18].

**Figure 3 jcm-12-05519-f003:**
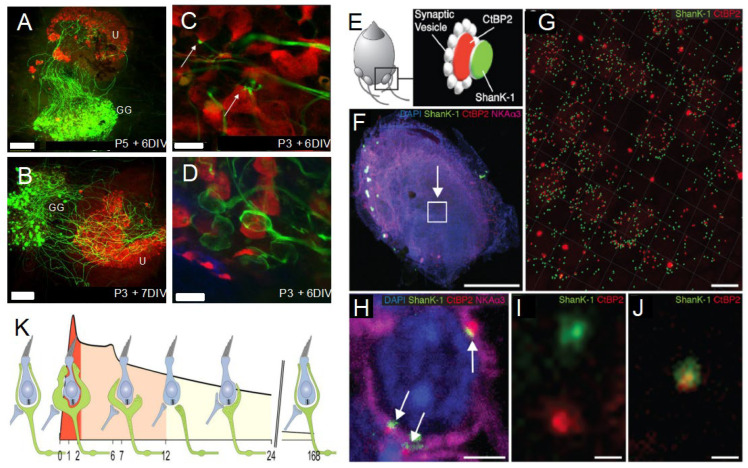
Spontaneous reafferentation of primary vestibular synapses. (**A**–**D**) Co-culture model of sensory epithelia and Scarpa’s ganglion. (**A**,**B**) Under culture conditions, the nerve processes regrow spontaneously from Scarpa’s ganglion (GG, green) towards the target organs—in this case, utricle (U, red). This regrowth is guided by the release of neurotrophin (BDNF) in the sensory epithelia. Scales bars 100 µm. Nerve fibres are able to form button (**C**) and calyx (**D**) contacts (arrows) with hair cells. These contacts display the characteristics of functional synapses. IVD: in vitro days; P: postnatal development day. Green: neurofilaments; red: calretinin. Scales bars 10 µm [24]. (**E**–**J**) Spontaneous repair of vestibular synapses in rodent with vestibular injury. (**E**). Illustration of immunocytochemical labelling of pre- and post-synaptic proteins in the adult mouse utricle. (**F**). Area of analysis in the utricle (arrow). (**G**). Immunolabelling in the test area. (**H**). Expression of pre- and post-synaptic proteins at a calyx synapse (arrows). Pre- (red) and post-synaptic proteins (green) in deafferented (**I**) and normal (**J**) conditions. (**K**) Illustration of the de-/reafferentation of vestibular hair cells occurring during the first week after injury (numbers: days post injury) (**E**–**J**) [20]; (**K**) [19].

**Figure 4 jcm-12-05519-f004:**
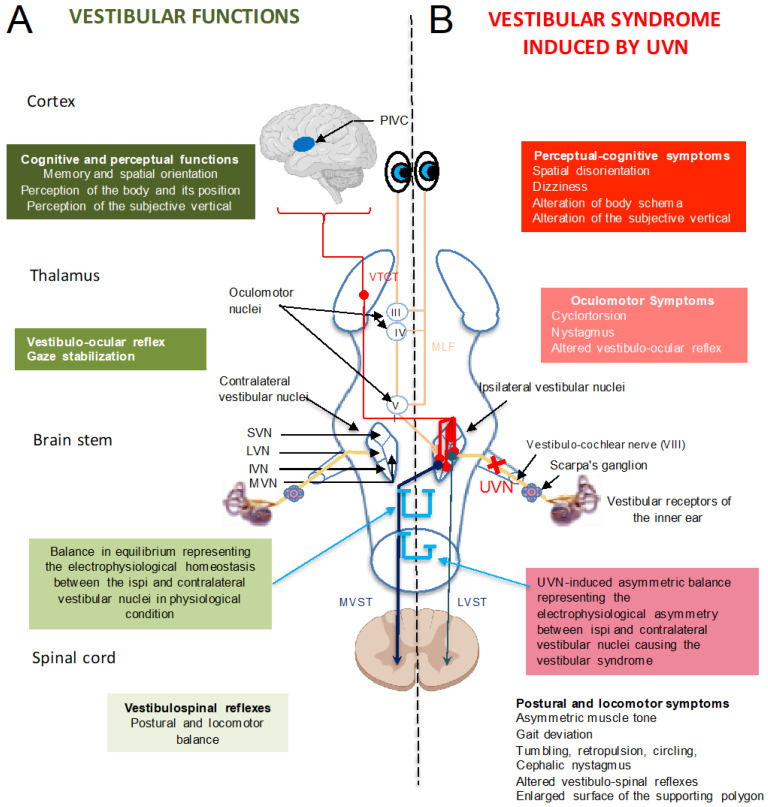
Illustration of the anatomo-functional organization of the vestibular nuclei complex (**A**) and the different components of the vestibular syndrome induced by unilateral loss of the peripheral vestibular organ (**B**). The vestibular nerve contacts the sensory cells of the peripheral labyrinthine receptors (semicircular canals, utricle and saccule) and projects ipsilaterally to four different vestibular nuclei (VN): medial (NVM), inferior (IVN), lateral (LVN) and superior (SVN). The VN are located in the dorso-lateral part of the bulboprotuberantial junction of the brainstem, under the floor of the fourth ventricle. They form the first relay of the vestibulo-ocular, vestibulo-spinal and vestibulo-vegetative reflexes and the ascending vestibulo-thalamo-cortical tract (VTCT) involved in perceptive and cognitive functions, such as spatial orientation. The VN are at the origin of premotor messages (in command of the ocular and somatic musculature) for the regulation of one’s posture and stabilization of one’s gaze. They project through the medial longitudinal fasciculus (MLF) on oculomotor nuclei, whose motoneurons control gaze stabilization during head movements. Vestibulo-spinal projections, originating from the ipsilateral LVN, reach all the medullary stages via the lateral vestibulo-spinal tract (LVST). Contralateral projections from the medial, inferior and lateral VN constitute the medial vestibulo-spinal tract (MVST) and control the motor neurons controlling the musculature of the neck and upper body axis.

**Figure 5 jcm-12-05519-f005:**
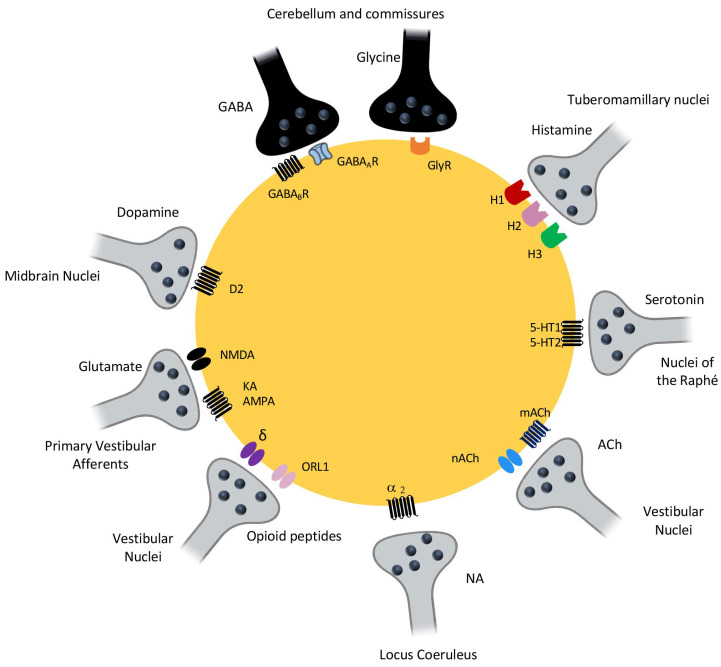
Illustration representing the diversity of neurotransmitter systems present in the vestibular nuclei. All neurochemical systems are present in the vestibular nuclei. The primary synaptic input to neurons in the vestibular nuclei is primary afferents from peripheral vestibular receptors. These pathways are mediated by glutamate, which interacts with NMDA, AMPA/KAINATE and metabotropic receptors. The vestibular nuclei also receive glutamatergic synapses from spinal cord neurons. GABAergic fibres originating mainly from the cerebellum and contralateral vestibular nuclei also converge on neurons in the vestibular nuclei and activate GABAa and GABAb receptors. Histaminergic fibres from the tuberomammillary nuclei of the posterior hypothalamus act on histaminergic H1, H2 and H3 receptors. Serotonergic fibres from the Raphé nuclei activate 5-HT1 and 5-HT2 receptors. Intrinsic and commissural connections give rise to glycinergic fibres acting on glycinergic inhibitory receptors. Noradrenergic fibres from the locus coeruleus act primarily on α2 receptors, but also on α1 and β adrenergic receptors. Dopaminergic fibres are also present in the VN and activate D2 receptors. Internuclear enkephalinergic fibres release opioid peptides acting on ORL1 receptors (orphan opioid receptor) and δ opioid receptors. Output from neurons in the vestibular nuclei is predominantly via glutamatergic and cholinergic projections, but GABAergic and glycinergic projections have also been demonstrated.

## Data Availability

Not applicable.

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
