# Peer review of "Cellular and Molecular Mechanisms of Vestibular Ageing"

_jcm, 2023, doi:10.3390/jcm12175519_

Round 1
Reviewer 1 Report
This is a very well-written review, with clear and well-defined topics on the subject, which will shed light on human aging, especially on the vestibular system and the motor skills regulated by this organ. Below are some suggestions for authors that I missed in the text and some suggestions for improving the text.
- Some topics contain the question mark, but they are not questions. Authors must reformulate the topics in questions, or if they are statements, remove the interrogations.
- Some paragraphs are very segmented, that is, there is no fluidity of the text, which connects what the authors are mentioning at the end of a paragraph, with the beginning of the next paragraph. I believe that authors can improve this connection and the fluidity between paragraphs.
- I expected to find in this review a greater discussion between the aging of the structures of the vestibular organs (otolithic canals and organs) with changes in postural stability and balance. However, the authors only mention that there may be alterations in these motor skills, I missed a greater explanation in this regard, if the authors could include it would be a very rich topic, which would encourage evidence and a model for Physical Therapists to justify their treatments in older adults with vestibulopathies.
There are many studies involving humans that correlate the aging of the auditory-vestibular systems with declines in human balance. Below I quote some evidence on the subject that may help authors in the construction or elaboration of this paragraph.
Bohannon RW, Larkin PA, Cook AC, Gear J, Singer J. Decrease in timed balance test scores with aging. Phys Ther. 1984; 64: 1067-70.
Laughton CA, Slavin M, Katdare K, Nolan L, Bean JF, Kerrigan DC, et al. Aging, muscle activity, and balance control: physiologic changes associated with balance impairment. Gait Posture. 2003; 18; 101-8.
Aslan UB, Cavlak U, Yagci N, Akdag B. Balance performance, aging and falling: a comparative study based on a Turkish sample. Arch Gerontol Geriatr. 2008; 46: 283-92.
Hatzitaki V, Amiridis IG, Arabatzi F. Aging effects on postural responses to self-imposed balance perturbations. Gait Posture. 2005; 22: 250-7.
Kanekar N, Aruin AS. Aging and balance control in response to external perturbations: role of anticipatory and compensatory postural mechanisms. Age. 2014; 36: 9621.
Pang BWJ, Wee SL, Lau LK, Jabbar KA, Seah WT, Ng DHM. Sensorimotor performance and reference values for fall risk assessment in community-dwelling adults: The Yishun study. Phys Ther. 2021; 101: 7: pzab035.
Freiberger E, Sieber CC, Kob R. Mobility in older community-dwelling persons: a narrative review. Front Physiol. 2020; 11: 881.
Nóbrega-Sousa P, Gobbi LTB, Orcioli-Silva D, Conceição NR, Beretta VS, Vitório R. Aging and associations with gait and executive function. Neurorehabil Neural Repair. 2020; 34: 915-24.
Osoba MY, Ashwini K, Agrawal SK, Lalwani AK. Balance and gait in the elderly: a contemporary review. Laryngoscope Investig Otolaryngol. 2019; 4: 143-53.
Quiao M, Feld JA, Franz JR. Aging effects on leg joint variability during walking with balance perturbations. Gait Posture. 2018; 62: 27-33.
Magnani RM, Bruijn SM, van Dieen JH, Vieira MF. Head orientation and gait stability in young adults, dancers and older adults. Gait Posture. 2020; 80: 68-73.
Porto JM, Iosimuta NCR, Freire-Junior RC, Braghin RMB, Leitner E, Freitas LG, et al. Risk factors for future falls among community-dwelling older adults without a fall in the previous year: a prospective one-year longitudinal study. Arch Gerontol Geriatr. 2020; 91: 104161.
Moscicki EK, Elkins EF, Baum HM, McNamara PM. Hearing loss in the elderly: an epidemiologic study of the Framinghham Heart Study Cohort. Ear Hear. 1985; 6: 184-90.
Ruwer SL, Rossi AG, Simon LF. Balance in the elderly. Braz J Otorhinolaryngol. 2005; 71: 298-303.
Su HC, Huang TW, Young YH, Cheng PW. Aging effect on vestibular evoked myogenic potential. Otol Neurotol. 2004; 25: 977-80.
Zalewski CK. Aging of the human vestibular system. Semin Hear. 2015; 36: 175-96.
Cuickshanks KJ, Wiley TL, Tweed TS, Klein BE, Mares-Perlman JA, Nondahl DM. Prevalence of hearing loss in older adults in Beaver Dam, Wisconsin. The epidemiology of hearing loss study. Am J Epidemiol. 1998; 48: 879-86.
Alexander TH, Harris JP. Incidence of sudden sensorineural hearing loss. Otol Neurotol. 2013; 34: 1586-9.
Huafeng Y, Hongqin W, Wenna Z, Yuan L, Peng X. Clinical characteristics and prognosis of elderly patients with idiopathic sudden sensorineural hearing loss. Acta Otolaryngol. 2019; 139: 866-9.
Thomas E, Martines F, Bianco A, Messina G, Giustino V, Zangla, et al. Decreased postural control in people with moderate hearing loss. Medicine (Baltimore). 2018; 97: e0244.
Ciquinato DSA, Doi MY, Silva RA, Oliveira MR, Gil AWO, Marchiori LLM. Posturographic analysis in the elderly with and without sensorineural hearing loss. Int Arch Otorhinolaryngol. 2020; 24: e496-502.
Bang S, Jeon J, Lee J, Choi J, Song J, Chae S. Association between hearing loss and postural instability in older Korean adults. JAMA Otolaryngol Head Neck Surg. 2020; 146: 530–4.
Carr S, Pichora-Fuller MK, Li KZH, Campos JL. Effects of age on listening and postural control during realistic multi-tasking conditions. Hum Mov Sci. 2020; 73: 102664.
Lin TC, Yen M, Liao YC. Hearing loss is a risk factor of disability in older adults: a systematic review. Arch Gerontol Geriatr. 2019; 85: 103907.
Sakurai R, Suzuki H, Ogawa S, Takahashi M, Fujiwara Y. Hearing loss and increased gait variability among older adults. Gait Posture. 2021; 87: 54-8.
Xu D, Newel MD, Francis AL. Fall-related injuries mediate the relationship between self-reported hearing loss and mortality in middle-aged and older adults. J Gerontol A Biol Sci Med Sci. 2021; 76: 213-20.
Szeto B, Zanotto D, Lopez EM, Stafford JA, Nemer JS, Chambers AR, et al. Hearing loss is associated with increase variability in double support period in the elderly. Sensors. 2021; 21: 278.
- The authors mention divergences between studies in animals and in humans, however, we must remember that the human body in relation to balance requires more coordination and motor control, when compared to animals, in view of the bipedal position adopted by humans. Which leads to a smaller base of support and a consequent greater intensity of muscle co-contractions to maintain achieving the desired posture, or maintaining the posture without causing changes, pain, or damage to the individual. This needs to be considered in this weighting between animal and human studies.
- ATTENTION: In several paragraphs the authors refer to the older adults as “older adults” and in some paragraphs as “elderly”. The term “elderly” is out of use, as it can be interpreted as a pejorative term. Therefore, authors should replace all terms “elderly” and standardize the term “older adults” throughout the manuscript.
- The authors mention that there are few studies on vestibular compensation, and cite a study with rats in the aquatic environment. However, I am aware of two studies that used aquatic physiotherapy exercises to improve vestibular outcomes in individuals with peripheral vestibular dysfunction. I cite the studies below, it may help to improve this paragraph.
Gabilan YPL, Perracini MR, Munhoz MSL, Ganança FF. Aquatic physiotherapy for vestibular rehabilitation in patients with unilateral vestibular hypofunction: exploratory prospective study. J Vestib Res. 2008; 18: 139-46.
Pereira CMM, Vale JSP, Oliveira WP, Pinto DS, Cal RVR, Azevedo YJ, et al. Aquatic physiotherapy: a vestibular rehabilitation option. Braz J Otorhinolaryngol. 2021; 87: 649-54.
- I know that this was not the objective of the study, however, I believe that the authors could end the study with a paragraph demonstrating how much vestibular rehabilitation exercises can contribute to improving vestibular function, even in the elderly. The literature contains several studies and systematic reviews, including those from Cochrane that may serve as evidence to support this paragraph.
Author Response
Comments and Suggestions for Authors
This is a very well-written review, with clear and well-defined topics on the subject, which will shed light on human aging, especially on the vestibular system and the motor skills regulated by this organ. Below are some suggestions for authors that I missed in the text and some suggestions for improving the text.
- Some topics contain the question mark, but they are not questions. Authors must reformulate the topics in questions, or if they are statements, remove the interrogations.
# To meet the reviewer requirement, we:
- removed the introducing sentence in line 42
- changed the sentence in line 89 from “What is the role of age-related synaptic loss in vestibular ageing?” to “Age-related synaptic loss and vestibular ageing”
- removed the sentences between lines 129-133 and modified the text between lines 134 and 140
- suppressed the question marks in lines 145 & 156
- Some paragraphs are very segmented, that is, there is no fluidity of the text, which connects what the authors are mentioning at the end of a paragraph, with the beginning of the next paragraph. I believe that authors can improve this connection and the fluidity between paragraphs.
# As suggested, we added changes throughout the text to smoothen the connection between subsequent paragraphs.
- I expected to find in this review a greater discussion between the aging of the structures of the vestibular organs (otolithic canals and organs) with changes in postural stability and balance. However, the authors only mention that there may be alterations in these motor skills, I missed a greater explanation in this regard, if the authors could include it would be a very rich topic, which would encourage evidence and a model for Physical Therapists to justify their treatments in older adults with vestibulopathies. There are many studies involving humans that correlate the aging of the auditory-vestibular systems with declines in human balance. Below I quote some evidence on the subject that may help authors in the construction or elaboration of this paragraph.
Bohannon RW, Larkin PA, Cook AC, Gear J, Singer J. Decrease in timed balance test scores with aging. Phys Ther. 1984; 64: 1067-70.
Laughton CA, Slavin M, Katdare K, Nolan L, Bean JF, Kerrigan DC, et al. Aging, muscle activity, and balance control: physiologic changes associated with balance impairment. Gait Posture. 2003; 18; 101-8.
Aslan UB, Cavlak U, Yagci N, Akdag B. Balance performance, aging and falling: a comparative study based on a Turkish sample. Arch Gerontol Geriatr. 2008; 46: 283-92.
Hatzitaki V, Amiridis IG, Arabatzi F. Aging effects on postural responses to self-imposed balance perturbations. Gait Posture. 2005; 22: 250-7.
Kanekar N, Aruin AS. Aging and balance control in response to external perturbations: role of anticipatory and compensatory postural mechanisms. Age. 2014; 36: 9621.
Pang BWJ, Wee SL, Lau LK, Jabbar KA, Seah WT, Ng DHM. Sensorimotor performance and reference values for fall risk assessment in community-dwelling adults: The Yishun study. Phys Ther. 2021; 101: 7: pzab035.
Freiberger E, Sieber CC, Kob R. Mobility in older community-dwelling persons: a narrative review. Front Physiol. 2020; 11: 881.
Nóbrega-Sousa P, Gobbi LTB, Orcioli-Silva D, Conceição NR, Beretta VS, Vitório R. Aging and associations with gait and executive function. Neurorehabil Neural Repair. 2020; 34: 915-24.
Osoba MY, Ashwini K, Agrawal SK, Lalwani AK. Balance and gait in the elderly: a contemporary review. Laryngoscope Investig Otolaryngol. 2019; 4: 143-53.
Quiao M, Feld JA, Franz JR. Aging effects on leg joint variability during walking with balance perturbations. Gait Posture. 2018; 62: 27-33.
Magnani RM, Bruijn SM, van Dieen JH, Vieira MF. Head orientation and gait stability in young adults, dancers and older adults. Gait Posture. 2020; 80: 68-73.
Porto JM, Iosimuta NCR, Freire-Junior RC, Braghin RMB, Leitner E, Freitas LG, et al. Risk factors for future falls among community-dwelling older adults without a fall in the previous year: a prospective one-year longitudinal study. Arch Gerontol Geriatr. 2020; 91: 104161.
Moscicki EK, Elkins EF, Baum HM, McNamara PM. Hearing loss in the elderly: an epidemiologic study of the Framinghham Heart Study Cohort. Ear Hear. 1985; 6: 184-90.
Ruwer SL, Rossi AG, Simon LF. Balance in the elderly. Braz J Otorhinolaryngol. 2005; 71: 298-303.
Su HC, Huang TW, Young YH, Cheng PW. Aging effect on vestibular evoked myogenic potential. Otol Neurotol. 2004; 25: 977-80.
Zalewski CK. Aging of the human vestibular system. Semin Hear. 2015; 36: 175-96.
Cuickshanks KJ, Wiley TL, Tweed TS, Klein BE, Mares-Perlman JA, Nondahl DM. Prevalence of hearing loss in older adults in Beaver Dam, Wisconsin. The epidemiology of hearing loss study. Am J Epidemiol. 1998; 48: 879-86.
Alexander TH, Harris JP. Incidence of sudden sensorineural hearing loss. Otol Neurotol. 2013; 34: 1586-9.
Huafeng Y, Hongqin W, Wenna Z, Yuan L, Peng X. Clinical characteristics and prognosis of elderly patients with idiopathic sudden sensorineural hearing loss. Acta Otolaryngol. 2019; 139: 866-9.
Thomas E, Martines F, Bianco A, Messina G, Giustino V, Zangla, et al. Decreased postural control in people with moderate hearing loss. Medicine (Baltimore). 2018; 97: e0244.
Ciquinato DSA, Doi MY, Silva RA, Oliveira MR, Gil AWO, Marchiori LLM. Posturographic analysis in the elderly with and without sensorineural hearing loss. Int Arch Otorhinolaryngol. 2020; 24: e496-502.
Bang S, Jeon J, Lee J, Choi J, Song J, Chae S. Association between hearing loss and postural instability in older Korean adults. JAMA Otolaryngol Head Neck Surg. 2020; 146: 530–4.
Carr S, Pichora-Fuller MK, Li KZH, Campos JL. Effects of age on listening and postural control during realistic multi-tasking conditions. Hum Mov Sci. 2020; 73: 102664.
Lin TC, Yen M, Liao YC. Hearing loss is a risk factor of disability in older adults: a systematic review. Arch Gerontol Geriatr. 2019; 85: 103907.
Sakurai R, Suzuki H, Ogawa S, Takahashi M, Fujiwara Y. Hearing loss and increased gait variability among older adults. Gait Posture. 2021; 87: 54-8.
Xu D, Newel MD, Francis AL. Fall-related injuries mediate the relationship between self-reported hearing loss and mortality in middle-aged and older adults. J Gerontol A Biol Sci Med Sci. 2021; 76: 213-20.
Szeto B, Zanotto D, Lopez EM, Stafford JA, Nemer JS, Chambers AR, et al. Hearing loss is associated with increase variability in double support period in the elderly. Sensors. 2021; 21: 278.
# Throughout this review, aimed at addressing the question of the cellular and molecular mechanisms of vestibular ageing, we have chosen to shed light specifically on age-dependent alterations which affect both the vestibular sensors and central vestibular pathways. We agree with the reviewer that enlarging the focus to the age-related changes affecting other sensory systems such as the vision or proprioception, as well as those affecting the musculoskeletal and tendon networks, visual integration pathways and adaptation mechanisms, metabolic functions and cognitive processes would have allowed to embrace the whole picture of the age-related posturo-locomotor deficits. However, this would deserve another project of special issue specifically devoted to this topic. Nevertheless, to follow the reviewer suggestion, we rewrote the Conclusion part to broadened the discussion beyond the purely vestibular sphere.
- The authors mention divergences between studies in animals and in humans, however, we must remember that the human body in relation to balance requires more coordination and motor control, when compared to animals, in view of the bipedal position adopted by humans. Which leads to a smaller base of support and a consequent greater intensity of muscle co-contractions to maintain achieving the desired posture, or maintaining the posture without causing changes, pain, or damage to the individual. This needs to be considered in this weighting between animal and human studies.
# We thank the reviewer for this very relevant comment. We added the sentences in our Conclusion.
- ATTENTION: In several paragraphs the authors refer to the older adults as “older adults” and in some paragraphs as “elderly”. The term “elderly” is out of use, as it can be interpreted as a pejorative term. Therefore, authors should replace all terms “elderly” and standardize the term “older adults” throughout the manuscript.
# Corrected throughout the text.
- The authors mention that there are few studies on vestibular compensation, and cite a study with rats in the aquatic environment. However, I am aware of two studies that used aquatic physiotherapy exercises to improve vestibular outcomes in individuals with peripheral vestibular dysfunction. I cite the studies below, it may help to improve this paragraph.
Gabilan YPL, Perracini MR, Munhoz MSL, Ganança FF. Aquatic physiotherapy for vestibular rehabilitation in patients with unilateral vestibular hypofunction: exploratory prospective study. J Vestib Res. 2008; 18: 139-46.
Pereira CMM, Vale JSP, Oliveira WP, Pinto DS, Cal RVR, Azevedo YJ, et al. Aquatic physiotherapy: a vestibular rehabilitation option. Braz J Otorhinolaryngol. 2021; 87: 649-54.
# We thank the reviewer for this comment. The cited studies in animal models of vestibular disorders were chosen as they could bring insights in the cellular and molecular mechanisms which support vestibular compensation which is the main subject of the review.
- I know that this was not the objective of the study, however, I believe that the authors could end the study with a paragraph demonstrating how much vestibular rehabilitation exercises can contribute to improving vestibular function, even in the elderly. The literature contains several studies and systematic reviews, including those from Cochrane that may serve as evidence to support this paragraph.
# In this review, we have chosen to shed light specifically on age-dependent alterations which affect both the vestibular sensors and central vestibular pathways. We agree with the reviewer that enlarging the focus to how much vestibular rehabilitation exercises can contribute to improving vestibular function would have enriched the discussion, however, we believe this topic deserves another project of special issue.

Reviewer 2 Report
The authors present a very thorough review of the cellular and molecular mechanisms involved in the aging process affecting the vestibular function, both analysing the deterioration of the vestibule and the adaptive mechanisms put in place in order to counteract the disturbances of the function. Several aspects are taken into account, delivering a very detailed description of the state of the art concerning vestibular function.
I really enjoyed the manuscript and I only have few minor suggestions that I think would improve the paper:
pag. 8, line 336: I guess it was meant 3 months old instead of 3 years old.
pag. 9: paragraph 3 and 4 feel out of place, I would suggest to link them more to the context of the review or remove them.
Figures 1 and 2: more detailed description in the legend would help understanding the figure by itself, without having to go back and look in the main text.
Figure 1B: font is barely readable, try to enlarge it, if needed.
Author Response
Comments and Suggestions for Authors
The authors present a very thorough review of the cellular and molecular mechanisms involved in the aging process affecting the vestibular function, both analyzing the deterioration of the vestibule and the adaptive mechanisms put in place in order to counteract the disturbances of the function. Several aspects are taken into account, delivering a very detailed description of the state of the art concerning vestibular function.
I really enjoyed the manuscript and I only have few minor suggestions that I think would improve the paper:
pag. 8, line 336: I guess it was meant 3 months old instead of 3 years old.
# Corrected
pag. 9: paragraph 3 and 4 feel out of place, I would suggest to link them more to the context of the review or remove them.
# To meet the reviewer suggestion we rewrote the paragraph 3 and 4 of the part 3.8 to improve reading.
Figures 1 and 2: more detailed description in the legend would help understanding the figure by itself, without having to go back and look in the main text.
# Legends of Fig 1 and Fig2 have been rewritten to improve reading.
Figure 1B: font is barely readable, try to enlarge it, if needed.
# Figure 1 has been corrected to improve reading.

Reviewer 3 Report
This paper reviews cellular and molecular mechanisms of vestibular ageing from two different perspectives. Focusing on the lesser-attended area of age-related vestibular sensory disorders, the article provides a systematic summary of the cellular and molecular aspects of what happens to the peripheral and central vestibular systems during aging. The authors summarize the molecular mechanisms of vestibular compensation in existing age-related vestibular dysfunction, pointing to the need to understand how vestibular damage in young adults governs the subsequent evolution of vestibular sensory networks. The aim is to identify targets for intervention to protect hair cells and subsequent vestibular neural pathways from damage. The paper is well written, and it could be further improved by considering following comments.
1. Are the panels cited in Figure authorized by the original author? The labeling in Figure2 is confusing, and some of them carry letter marks from the references. This is not standardized. Please contact the author for the original figure.
2. Lacking a general INTRODUCTION, please discuss the reasons for writing according to the categorization of the peripheral vs. central vestibular system? What vestibular structures are included in the peripheral and central, please add more details.
3. In Part 1.1, are there any studies of vestibular hair cell loss and mechanisms in animal models of aging? Is it possible to obtain some evidence of damage phenotypes and molecular mechanisms from animal models?
4. In Part 1.2, the description of synaptic loss is interspersed with hair cell loss, which is confusing. Please reframe the description of each part.
5. Lines152-155 of Part 1.3. The content of this section is subjective speculation about vestibular phenotypes based on age-related deafness pathologic phenotypes and is not persuasive, please find more evidence.
6. The structure of the paper is a bit disorganized. There is no continuity in each part of the content, and they are independent from each other. Moreover, there is no good logical relationship between the paragraphs, which can refer to the order of phenotypic observation-molecular mechanism-rescue and repair, and so on.
7. The lower text in the A-D panels of Figure 7 is not neatly labeled. Please correct it.
8. Line 614-628, please modify this section to suit your own situation.
Minor editing of English language required.
Author Response
Comments and Suggestions for Authors
This paper reviews cellular and molecular mechanisms of vestibular ageing from two different perspectives. Focusing on the lesser-attended area of age-related vestibular sensory disorders, the article provides a systematic summary of the cellular and molecular aspects of what happens to the peripheral and central vestibular systems during aging. The authors summarize the molecular mechanisms of vestibular compensation in existing age-related vestibular dysfunction, pointing to the need to understand how vestibular damage in young adults governs the subsequent evolution of vestibular sensory networks. The aim is to identify targets for intervention to protect hair cells and subsequent vestibular neural pathways from damage. The paper is well written, and it could be further improved by considering following comments.
- Are the panels cited in Figure authorized by the original author? The labeling in Figure2 is confusing, and some of them carry letter marks from the references. This is not standardized. Please contact the author for the original figure.
# Panels from Figure 2 have been taken from reference [18], published in Frontiers of Aging Neuroscience in 2019. All Frontiers articles from July 2012 onwards are published with open access under the Creative Commons CC-BY license (the current version is CC-BY, version 4.0). This means that the author(s) retains copyright, but the content is free to download, distribute, and adapt for commercial or non-commercial purposes, given appropriate attribution to the original article. We propose to address the question of the letter marks with the JCM publication office.
- Lacking a general INTRODUCTION, please discuss the reasons for writing according to the categorization of the peripheral vs. central vestibular system? What vestibular structures are included in the peripheral and central, please add more details.
# As suggested by the reviewer, we added an introductive paragraph to discuss why we have chosen to shed light specifically on age-dependent alterations which affect both the vestibular sensors and central vestibular pathways. We also repositioned this choice among the set of other changes involved in the age-related posturo-locomotor deficits. We also modified the review abstract.
- In Part 1.1, are there any studies of vestibular hair cell loss and mechanisms in animal models of aging? Is it possible to obtain some evidence of damage phenotypes and molecular mechanisms from animal models?
# To our knowledge, there are no studies of hair cell loss and mechanisms in animal models of aging. However, rodent models of ageing, in particular genetically modified mouse models, have provided a better understanding of the impact of molecular changes on the development of age-related vestibular deficits. We have chosen to add a paragraph dedicated to this question in 2.6, citing two recent reviews.
- In Part 1.2, the description of synaptic loss is interspersed with hair cell loss, which is confusing. Please reframe the description of each part.
# To meet the reviewer comment we simplified this paragraph by removing the text regarding the hair cell loss in the cochlea, that we believe was a bit out of focus. We corrected the references accordingly.
- Lines152-155 of Part 1.3. The content of this section is subjective speculation about vestibular phenotypes based on age-related deafness pathologic phenotypes and is not persuasive, please find more evidence.
# We agree with the reviewer that this statement is purely speculative. However based on the similarities between auditory and vestibular primary synapses physiology and pathophysiology, the possibility of accelerated age-related synaptic loss in previously damaged/deafferented vestibular primary synapses cannot be discarded. We corrected the sentence accordingly.
- The structure of the paper is a bit disorganized. There is no continuity in each part of the content, and they are independent from each other. Moreover, there is no good logical relationship between the paragraphs, which can refer to the order of phenotypic observation-molecular mechanism-rescue and repair, and so on.
# Throughout this review, aimed at addressing the question of the cellular and molecular mechanisms of vestibular ageing, we have chosen to shed light specifically on age-dependent alterations which affect both the vestibular sensors and central vestibular pathways. The paper is organized in two mains parts, and some parts contain sub-paragraphs we believe are well identified. To meet the reviewer suggestion we however rewrote the 2.6 and 2.8 part to add links between paragraphs and improve reading.
- The lower text in the A-D panels of Figure 7 is not neatly labeled. Please correct it.
# We suppose the reviewer is referring to Figure 3. The lower text in the A-D panel has been corrected to improve reading.
- Line 614-628, please modify this section to suit your own situation.
# We have modified this section slightly. In this part of the article, we wanted to talk about our expertise in vestibular compensation in young adults and extrapolate it to the elderly. Indeed, if we consider that aging, although non-pathological, produces the same effects as vestibular injury, we might expect the aging vestibular tissue to set up the same adaptive plastic processes to maintain operational vestibular function. Similarly, in the event of a vestibular lesion in the elderly, these same plastic processes, although diminished by age, could come into play to restore and maintain vestibular function. Knowledge of these plasticity processes is therefore important for the therapeutic management of vestibular function in the elderly.

Round 2
Reviewer 3 Report
The article systematically reviewed the cellular and molecular mechanisms of vestibular ageing from both peripheral and central vestibular system aspects and pointed out that any interventional action must first be based on a real understanding of the critical periods present during the development of the vestibular sensory networks. The article has chosen an interesting topic.
There are a few revision suggestions to consider before the article is published.
1. Lines 25-38 of the INTRODUCTION are not labeled with citations, so please double-check to see if you have cited any of the relevant articles.
2. Lines 571-590 are more of a summary of the entire article than a summary of the topic "Re-emergence of a critical post-injury period reproducing developmental plasticity processes ".
3. The conclusion in lines 591-613 lacks organization. In the CONCLUSION section, the authors could consider a brief and systematic summary of the studies mentioned in the article.
4. Please make sure that the title of the article in lines 707-708 is correct.
Minor editing of English language required
Author Response
There are a few revision suggestions to consider before the article is published.
- Lines 25-38 of the INTRODUCTION are not labeled with citations, so please double-check to see if you have cited any of the relevant articles.
# The mentioned lines are introductive to subsequent paragraphs which we believe are properly referenced to relevant articles.
- Lines 571-590 are more of a summary of the entire article than a summary of the topic "Re-emergence of a critical post-injury period reproducing developmental plasticity processes ".
# We agree with this comment. We moved these lines to a new paragraph (3.9).
- The conclusion in lines 591-613 lacks organization. In the CONCLUSION section, the authors could consider a brief and systematic summary of the studies mentioned in the article.
# We understand the reviewer suggestion. However, we prefer to devote the conclusion part, first to remind the discrepancy between animal models and humans as suggested by another reviewer and second, to evoke the other contributing factors to age-related vestibula deficits.
- Please make sure that the title of the article in lines 707-708 is correct.
# We corrected the reference Igarashi et al.